# GCN5 Is a Master Regulator of Gene Expression in the Malaria Parasite *Plasmodium falciparum*

**DOI:** 10.3390/cells14120876

**Published:** 2025-06-10

**Authors:** Amuza Byaruhanga Lucky, Ahmad Rushdi Shakri, Xiaoying Liang, Hui Min, Xiao-Lian Li, Swamy Rakesh Adapa, Rays H. Y. Jiang, Liwang Cui, Chengqi Wang, Jun Miao

**Affiliations:** 1Department of Internal Medicine, Morsani College of Medicine, University of South Florida, 3720 Spectrum Blvd, Tampa, FL 33612, USA; abl@usf.edu (A.B.L.); arshakri@usf.edu (A.R.S.); xiaoying.liang@moffitt.org (X.L.); hmin@cmu.edu.cn (H.M.); xli16@usf.edu (X.-L.L.); liwangcui@usf.edu (L.C.); 2Center for Global Health and Infectious Diseases Research, College of Public Health, University of South Florida, 3720 Spectrum Blvd, Tampa, FL 33612, USA; swamyrakesh@usf.edu (S.R.A.); jiang2@usf.edu (R.H.Y.J.)

**Keywords:** histone acetyltransferase, GCN5, transcription factor, gene regulation, malaria, *Plasmodium falciparum*

## Abstract

GCN5-containing SAGA complex is evolutionarily conserved across yeast, plants, and humans and acts as a general transcription coactivator in the genome-wide regulation of genes. In *Plasmodium falciparum*, PfGCN5 forms a divergent complex, and the mis-localization of this complex by deleting the PfGCN5 bromodomain (ΔBrd) causes a plethora of growth defects. To directly test the PfGCN5 function, we performed conditional knockdown (KD) of *PfGCN5*. Whereas *PfGCN5* KD phenotypically recapitulated the ΔBrd growth defects, it caused fewer transcriptional alterations compared to ΔBrd. To decipher the mechanism by which PfGCN5 regulates gene expression, we applied a new chromatin landscape analysis tool, CUT&Tag-seq, to map the chromatin localization of PfGCN5 and its deposited histone mark H3K9ac. Compared to ChIP-seq, CUT&Tag-seq identified substantially more H3K9ac peaks in the promoters of its target genes, with the peak intensity positively correlated with the levels of gene expression. CUT&Tag-seq analysis was remarkably more sensitive in mapping chromatin positions of PfGCN5, which colocalized with H3K9ac. The genes enriched with PfGCN5/H3K9ac signals at their promoters are involved in broad biological processes. Notably, PfGCN5′s positions overlapped with sequence motifs recognized by multiple apetela2 (AP2)-domain-containing transcription factors (AP2 TFs), suggesting that they recruited PfGCN5 to these promoters. Additionally, PfGCN5 was also colocalized with AP2-LT, further validating that AP2-LT is an integral component of the PfGCN5 complex. Collectively, these findings establish PfGCN5 as a master gene regulator in controlling general and parasite-specific cellular processes in this low-branching parasitic protist.

## 1. Introduction

Histone Post-translational modification (PTM) is a key mechanism for regulating chromatin structure, affecting essential cellular processes of transcription, DNA replication, and DNA repair [1]. Histone acetylation, catalyzed by the histone acetyltransferases (HATs), is correlated with active transcription [2]. GCN5, a major HAT, exists in the SAGA (Spt-Ada-Gcn5 acetyltransferase) complex, which is evolutionarily conserved in model eukaryotes [3,4]. Earlier reports show that SAGA is involved in the regulation of ~10% of yeast and plant genes [5,6], whereas it was reported later that yeast SAGA was localized in all promoters, and its disruption downregulated each gene [7,8]. These findings indicate that SAGA acts as a general coactivator and promotes RNA polymerase II transcription via recruiting the preinitiation complex [9]. Specific transcription factors (TFs) bind motifs in the promoters of target genes and recruit the SAGA complex for gene activation. In plants, for example, the SAGA complex is recruited to the response genes under cold stress conditions by specific AP2-domain TFs [10,11]. In yeast, starvation-specific TFs recruit the SAGA complex to activate gluconeogenic and fat metabolism genes after glucose starvation [12].

*Plasmodium falciparum*, belonging to an early-branching eukaryote, caused about 597,000 deaths in 2023 alone [13]. Its complicated lifecycle, switching between vertebrate and mosquito, needs precise transcriptional regulation to deal with the rapid growth and substantial environmental changes in the host [14,15,16]. However, *Plasmodium* has a deficiency in specific TFs; only ~30 TFs, including 27 apicomplexan-specific AP2 TFs, have been identified so far [17,18,19,20,21], whereas *Saccharomyces cerevisiae* has almost the same size of genome and contains ~170 specific TFs [22]. In sharp contrast, many proteins involved in chromatin biology have been identified, highlighting the significance of epigenetic regulation in malaria parasites [21,23,24]. *P. falciparum* chromatin mainly consists of euchromatin, with heterochromatic regions restricted to sub-telomeres and a few internal loci [25,26,27,28,29,30,31]. These spars of heterochromatin are clustered at the perinucleus and decorated by the histone H3K9me3 modification and heterochromatin protein 1 (HP1), which regulate antigenic variation, gametocytogenesis, and drug sensitivity [25,26,31,32,33,34,35,36]. Euchromatin is marked with the histone markers H3K9ac, H3K4me3, and H4K8ac [26,37,38,39], probably created by the two major HATs (PfGCN5 and PfMYST) and SET1, a histone methyltransferase, respectively [40,41,42,43]. Notably, H3K9ac, but not H3K4me3, correlates well with the transcriptional status of the genes [38]. Despite the significance of these histone PTMs in modulating gene activities, how they act during parasite development is still unclear [23,24,44,45,46,47,48,49].

The GCN5 complex in malaria parasites has diverged significantly from evolutionarily conserved SAGA complexes. Efforts to identify readers of the PTMs and PfGCN5-associated proteins in *P. falciparum* identified a PfGCN5 complex containing nine subunits. This PfGCN5 complex only shares two subunits with the SAGA complex (namely PfGCN5 and PfADA2) and contains several novel subunits, such as two plant homeodomain (PHD)-containing proteins (PfPHD1 and PfPHD2) and an AP2 TF (AP2-LT) [50,51]. Within this complex, PfGCN5 is the catalytic subunit, and its HAT domain preferentially acetylates histone H3 at K9 and K14 in vitro [40]. The bromodomain (Brd) in GCN5 binds the acetylated H3 tail, and the PHD in PfPHD1 binds H3K4me2/3, facilitating the recruitment and anchoring of the complex to euchromatin [50,52]. Both *PfGCN5 and PfPHD1* are essential for asexual growth; deletions of either PfGCN5 Brd or PfPHD1 PHD led to reduced merozoite invasion of erythrocytes and the disruption of mutually exclusive transcription of *var* genes [51]. Both domain deletion mutants showed the downregulation of invasion-related genes and upregulation of heterochromatic genes such as *var* genes [51]. Consistently, levels of euchromatin mark H3K9ac and H3K4me3 were markedly reduced in both deletion mutants, results that mimic those from chemical inhibition of the PfGCN5 activity [41,53]. Our recent PfGCN5 knockdown (KD) analysis further confirmed the PfGCN5 involvement in regulating stress responses [54]. While these studies have demonstrated that the PfGCN5 complex plays an essential function in the regulation of critical parasite processes, the mechanistic details remain to be determined.

One key approach to gaining a mechanistic understanding of chromatin regulation is to determine the chromatin occupancy profile of epigenetic factors. Recent Chromatin ImmunoPrecipitation (ChIP)-seq studies detected PfGCN5 enrichment at a small number of genes, in contrast to the broad distribution of H3K9ac, an active mark deposited by PfGCN5 [41,55]. This highlights the challenges of ChIP-seq in studying the chromatin landscape of coactivators such as GCN5, given their weak and transient chromatin binding activity [8]. Recently, a more sensitive method, called Cleavage Under Targets and Tagmentation-sequencing (CUT&Tag-seq) (https://www.activemotif.com/blog-cut-tag), was developed using the fusion protein consisting of Protein A and Tn5 transposase (pA-Tn5) preloaded with sequencing adapters. Upon activation, Tn5 performs targeted tagmentation, and the genomic fragments are ready for sequencing after PCR enrichment [56,57]. CUT&Tag-seq outperforms ChIP-seq in terms of sensitivity, resolution, and background noise, while it requires substantially lower cell inputs and sequencing depth [56,57]. Using this robust tool, we recently successfully determined the chromatin profile of H3R2me2 in *P. falciparum* [58].

In this study, we further delved into the PfGCN5 function by characterizing the conditional KD of PfGCN5 and its corresponding chromatic signals. While our results further validated our earlier findings from PfGCN5 and PfPHD1 domain deletions, we provide more details of PfGCN5’s recruitment to specific chromatin regions by performing chromatin landscape analysis using the CUT&Tag-seq technology.

## 2. Results

### 2.1. PfGCN5 KD Results in Growth Defects

Our earlier study showed that PfGCN5 Brd deletion (∆Brd) led to growth defects primarily due to the substantially lower capacity of merozoite invasion, accompanied by significant decreases in H3K9ac and H3K4me3 levels and downregulation of invasion-related genes [51]. In addition, ∆Brd also led to the loss of mutually exclusive expression patterns of *var* family genes, leading to an overall upregulation of all *var* genes [51]. Having recognized that the effects of ∆Brd presumably attributed to the disruption of PfGCN5′s binding to H3K9ac and the resultant mislocalization of the PfGCN5 complex, which may not faithfully reveal the PfGCN5 functions, we applied the TetR-DOZI conditional KD system to reduce PfGCN5 expression [54]. PfGCN5 protein expression was reduced to ~one-half after withdrawal of anhydrous tetracycline (aTc), resulting in significantly slower growth of the parasites [54]. To comprehensively characterize the phenotypes, we measured the proliferation rates and the progression of the intraerythrocytic developmental cycle (IDC). PfGCN5 KD (−aTc) parasites had a much lower proliferation rate of 2.6 compared to 4.3 in the +aTc parasites (Figure 1A, *p <* 0.01, unpaired *t*-test). Since IDC progression was not altered upon PfGCN5 KD (Figure 1B), we wanted to determine whether the growth defects were in the invasion process. The merozoite number in both −aTc and +aTc were similar (Figure 1C), but the invasion capacities of −aTc merozoites were reduced to ~50% (Figure 1D, *p <* 0.01, unpaired *t*-test). Additionally, KD parasites (−aTc) exhibited higher gametocytemia than the +aTc parasites (Figure 1E, *p <* 0.05, unpaired *t*-test). Taken together, KD of PfGCN5 resulted in growth phenotypes similar to those observed from PfGCN5 ∆Brd, except that ∆Brd also caused a 2–3 h extension of the ring stage [51].

### 2.2. PfGCN5 KD Led to Globally Reduced H3K9ac and H3K4me3 Levels

To determine whether PfGCN5 KD causes any changes in euchromatic histone PTMs, histones in the ring, trophozoite, and schizont stage parasites were purified for Western blots. H3K9ac was significantly reduced upon PfGCN5 KD, especially at the trophozoite stage when PfGCN5 expression is at its peak, consistent with the fact that PfGCN5 preferentially acetylates H3 at K9 in vitro [40] (Figure 1F). In contrast, tetra-acetylation of H4 (H4acs), which is known to be catalyzed by PfMYST [42], was not altered upon PfGCN5 KD. Furthermore, PfGCN5 KD also led to a ~30% reduction in the H3K4me3 level at the trophozoite stage (Figure 1F), which is specifically bound by PfPHD1. Similarly, GCN5 deletion also reduced the levels of both H3K9ac and H3K4me3 in model organisms [7,8]. Collectively, these results echo similar changes on the euchromatin histone marks after PfGCN5 ∆Brd, confirming PfGCN5′s vital role in histone H3 acetylation and the existence of crosstalk between two euchromatin marks (H3K9ac and H3K4me3).

### 2.3. PfGCN5 KD Profoundly Affects Global Transcription and Specific Cellular Pathways

By bridging TF and the transcription preinitiation complex (PIC), the SAGA complex facilitates the target genes’ transcription; the knockout of GCN5 and subunits of the SAGA complex led to a general reduction in transcriptional activation [7,8]. To determine how PfGCN5 KD disturbs the overall transcription, we compared the transcriptomes between −aTc and +aTc parasites during IDC by RNA-seq (Appendix A). As expected, the *PfGCN5* mRNA levels were reduced in all stages in −aTc parasites (Figure 2A, Appendix A). PfGCN5 KD profoundly disturbed the transcriptomes, with 2014 (35.7%) genes to be differentially expressed in at least one time-point analyzed (Appendix A). Specifically, 172, 403, 591, and 487 genes were downregulated, and 30, 387, 497, and 554 genes were upregulated in the rings, early trophozoites, late trophozoites, and schizonts, respectively (Figure 2B–F). Notably, a similar number of genes were up-and downregulated in all developmental stages except the ring stage, where the downregulated genes were ~5.7-fold higher than the upregulated ones. Overall, PfGCN5 ∆Brd more profoundly disturbed the transcriptomes, with a much higher number of differentially expressed genes (3533) during the IDC (Figure 2B right panel) [51]. Moreover, differentially expressed genes in PfGCN5 KD showed limited overlaps with those in PfGCN5 ∆Brd at the ring and early trophozoite stages, whereas 55.3%, 25.5% of upregulated and 64.1%, 27.7% of downregulated genes were shared between PfGCN5 KD and ∆Brd at the late trophozoite and schizont stages, respectively (Figure 2G, Appendix A). These shared genes include invasion-related genes that were downregulated in both PfGCN5 KD and ∆Brd. Collectively, these results showed that PfGCN5 KD caused marked transcriptomic changes, with similar trends in the later stages but relatively large variations at the early stages compared to the PfGCN5 ∆Brd.

*Gene Set Enrichment Analysis* (*GSEA*) [59,60] showed that in early trophozoites, genes associated with merozoite invasion, virulence/cytoadherence, the structure of telomere/sub-telomere, and genes encoding exported proteins and proteins containing phosphorylation and S-glutathionylation modifications were upregulated (Figure 2H, Appendix A). In late trophozoites and schizonts, genes associated with virulence (e.g., *var* family genes), telomere/sub-telomere structures, and genes encoding exported proteins were upregulated, while genes associated with merozoite invasion, protein phosphorylation, and S-glutathionylation were downregulated. Of the 86 putative invasion-related genes [61], 72 genes were downregulated in PfGCN5 KD parasites at the late trophozoite stage (Figure 2I), consistent with the reduced RBC invasion phenotype (Figure 1D). Genes encoding certain chromatin remodeling complexes and histone PTM readers were downregulated throughout the IDC (Figure 2H, Appendix A).

Compared to *GSEA* analyses of PfGCN5 ∆Brd [51], both KD and ∆Brd altered transcript levels in the genes related to merozoite invasion, virulence, exported proteins, telomere/sub-telomere structures, and PTMs. However, PfGCN5 ∆Brd also led to changes in additional pathways (e.g., cell cycle, DNA replication, translation, and gametocyte), indicating that ~50% reduction of PfGCN5 by KD resulted in transcriptional changes in parasite-specific pathways, such as host cell invasion and pathogenesis, whereas PfGCN5 ∆Brd also caused more extensive changes in the general cellular pathways, such as gene transcription and protein translation.

### 2.4. PfGCN5 KD Disturbs the Transcription of AP2 TFs

Our previous study showed that PfGCN5 ∆Brd led to substantial changes in the transcription of AP2 TFs [51]. Likewise, PfGCN5 KD also caused substantial transcriptional changes in all 27 ApiAP2 TFs (Figure 3A). The levels of certain AP2 TFs involved in telomere and heterochromatin maintenance and regulation (PfSIP2, PfAP2Tel, and PfAP2-heterochromatin-associated factors: PfAP2-HC, PfAP2-G2, PfAP2-G5, PfAP2-O2, PfAP2-O5, PfAP2-exp, PfAP2-11A, and PfAP2-12) [62,63,64,65] were disturbed upon PfGCN5 KD (Figure 3A). These changes may de-repress heterochromatic genes including var family genes. Some AP2 TFs known to regulate merozoite invasion (AP2-I, AP2-G, and PfAP2-11A, also called AP2-P) [66,67,68] were upregulated upon PfGCN5 KD, probably to compensate for the invasion defects caused by PfGCN5 KD. AP2-G, the master regulator of gametocytogenesis [68,69], was substantially upregulated upon PfGCN5 KD (Figure 3A), consistent with the increased gametocytogenesis in PfGCN5 KD parasites (Figure 1E). Interestingly, AP2-LT, a subunit of the PfGCN5 complex [51], was significantly downregulated at the late trophozoite stage, coinciding with its peak expression alongside PfGCN5 in the wildtype (WT) parasite.

### 2.5. PfGCN5 KD Broadly Alters Chromatin Structures

Our previous study showed that the downregulated genes after *PfGCN5* ∆Brd were the genes normally localized in the open, euchromatin regions in the WT parasites [51]. On the other hand, the upregulated genes after *PfGCN5* ∆Brd are those frequently associated with heterochromatin loci in the WT parasites. All these suggest that *PfGCN5* ∆Brd caused euchromatin to be less accessible to those downregulated genes and heterochromatin to be more accessible to those upregulated genes [51]. To determine the chromatin status upon PfGCN5 KD, we analyzed the chromatin accessibility and heterochromatic signals of differentially expressed genes using the ATAC-seq and HP1 ChIP-seq data [24,70]. At all time points, the downregulated genes are more open than the upregulated genes in the corresponding promoters in the WT parasites (Figure 3B), whereas upregulated genes upon PfGCN5 KD are more associated with the heterochromatin loci only at the schizont stage (Figure 3C). This indicates that PfGCN5 KD affected chromatin structures similarly, but to a relatively lesser extent, than *PfGNC5* ΔBrd.

### 2.6. CUT&Tag-seq Is More Sensitive for Profiling Histone PTMs and “Writers”

To more accurately map the histone PTMs and their “writers”, we first checked whether the CUT&Tag-seq is more sensitive than ChIP-seq. For mapping H3K9ac by ChIP-seq, 1 × 10^8^ 3D7 schizonts were used, and the H3K9ac profile obtained from 5 million reads was highly correlated (R = 0.62) with the published H3K9ac ChIP-seq data [38]. However, the published H3K9ac ChIP-seq failed to identify H3K9ac occupancy peaks, and our ChIP-seq data only identified five peaks (Figure 4A,B), although the coverage of H3K9ac signals in the promoters was positively correlated with gene expression (Figure 4C, R = 0.019). Additionally, H3K9ac signals were negatively correlated with the occupancy of H3K9me3, a silent mark in heterochromatin [71] (Figure 4A,B). For the CUT&Tag-seq procedure, with a ~200-fold lower number of schizonts and ~20-fold lower sequencing reads, we obtained highly reproducible results from two replicates (R = 0.81). CUT&Tag-seq resulted in a very low background from the IgG control, allowing us to identify 1869 H3K9ac peaks (Figure 4A,B). The positive correlation between the H3K9ac signals and gene expression was more robust (Figure 4D, R = 0.12).

Next, we conducted CUT&Tag-seq to map PfGCN5 using the anti-GFP antibodies in the PfGCN5::GFP parasite line at the schizont stage. Compared to the 3D7 control, we identified 986 PfGCN5 occupancy peaks (Figure 4B). Notably, the PfGCN5 peaks were highly colocalized with H3K9ac signals (Figure 4E) and positively correlated with gene expression (Figure 4F). This work highlights the better suitability of CUT&Tag-seq for mapping histone PTMs and their “writers” such as PfGCN5.

### 2.7. Extensive Regulation by Enrichment of H3K9ac/PfGCN5 at Promoters

In the schizont stage, 231 (23.4%) of PfGCN5 peaks and 565 (30.2%) of H3K9ac peaks were localized in 5′UTRs (Figure 5A,B). GO enrichment analysis of the 284 downstream neighboring genes with 5′ UTR PfGCN5 peaks showed that PfGCN5-enriched genes are involved in various biological processes including merozoite invasion (rhoptry, apical complex, inner membrane complex, and anchored components on merozoite membrane), transcriptional regulation, actin filament organization, protein phosphorylation, nuclear chromosome organization such as nucleosome and chromatin silencing (e.g., Sir2A), and vesicle trafficking (Figure 5C). These genes also include eight AP2 TFs (AP2-O2, AP2-P, PF3D7_0613800, AP2-L, SIP2, AP2-O, AP2-EXP, and AP2-G) and three Kelch13 interaction candidates (KICs) (KIC3, 4, and 8) that are known to regulate hemoglobin uptake. In contrast, ~2.5-fold more genes (699 vs. 284) were enriched with H3K9ac peaks at their 5′ UTRs compared to the PfGCN5-enriched genes, including AP2 TFs (AP2-12, AP2-P, AP2-FG/G3, PF3D7_1305200, AP2-O5, and AP2-O4), two KICs (KIC3, 4), GCN5, and HP1. Compared to PfGCN5-enriched genes, GO enrichment analysis showed that H3K9ac-enriched genes include similar GO terms such as merozoite invasion (apical complex, microneme), transcriptional regulation, actin filament organization, and vesicle trafficking), whereas they also include more GO terms such as regulation of microtubule-based movement, basal complex, mitochondrial transmembrane transporter activity, and protein ubiquitination (Appendix A). Interestingly, PfGCN5-enriched genes were significantly downregulated in the PfGCN5 ΔBrd parasites but not in PfGCN5 KD parasites, probably because ΔBrd caused more extensive transcriptomic changes than PfGCN5 KD (Figure 2B and Figure 5D, *p* < 0.001 Wilcoxon rank-sum test, Appendix A). Furthermore, PfGCN5 and H3K9ac signals were highly colocalized with ATAC-seq signals and an active histone mark H2A.Z but weakly colocalized with another active mark H3K4me3, further indicating that these peaks are in the open chromatin regions and involved in gene activation (Figure 5E,F). Additionally, 733 (74.3%) PfGCN5 peaks and 1247 (66.7%) H3K9ac peaks were localized in exons, consistent with the function of GCN5 in the regulation of transcriptional extension (Figure 5A,B).

### 2.8. The PfGCN5 Complex Is Potentially Recruited by Specific AP2 TFs

The presence of AP2 TFs in the PfGCN5 interactome suggests the coordination interaction between AP2 TFs and the PfGCN5 complex in gene regulation [50,51]. Using DREME for motif enrichment analysis [72], we identified several motifs within PfGCN5 chromatin occupancy sites (peaks) that overlap with fifteen DNA binding motifs of multiple TFs, including HDP1 and 12 AP2-domaining containing proteins, consisting of AP2-I, AP2-P, AP2-LT, AP2-L, Pf3D70613800, and seven heterochromatin-associated AP2s (AP2-12, AP2-HC, AP2-EXP, SIP2, AP2-G5, AP2-O2, and AP2-O5). These TFs are known to be involved in the regulation of invasion, antigenic variation (*var* genes), and sexual development (e.g., gametocytogenesis) [20,65,66,67] (Figure 6A and Appendix A). Similar motifs were identified from the top 5% PfGCN5 coverage in 5′UTRs (Figure 6B and Appendix A). From the H3K9ac 5′ UTR signals (H3K9ac peaks or top 5% H3K9ac coverage), additional motifs were identified, including most of the PfGCN5-enriched motifs (Appendix A). A recent report showed that AP2-LT is not recruited to its target chromatin sites by its binding motif, which was identified by in vitro protein-binding microarrays (PBMs), but rather by a motif similar to those of AP2-I, AP2-P, SIP, and HDP1 [73]. Notably, PfGCN5 chromatic binding sites colocalized with the recently published AP2-LT signals using ChIP-seq [65,73] (Figure 6C), further supporting AP2-LT as an intrinsic subunit of the PfGCN5 complex. Furthermore, PfGCN5 signals were also colocalized with those of AP2-I and AP2-P/AP2-11A [65,66,67,68] (Figure 6D,E and Appendix A). Collectively, these findings suggest that PfGCN5 is recruited to 5′UTR of its target genes by specific TFs to regulate different biological pathways.

## 3. Discussion

Building on our recent work on PfGCN5, including the identification of a unique PfGCN5 complex and functional characterization of the PfGCN5 complex’s two major subunits (PfGCN5 and PfPHD1) through domain deletions and stress responses [50,51,54], we further characterized PfGCN5 function by direct gene KD and chromatin landscape profiling in this study. Our results not only confirmed the critical role of PfGCN5 in regulating a large number of genes but also uncovered the potential mechanism of the PfGCN5 complex-mediated transcriptional regulation via coordination with the parasite-specific TFs.

Conditional KD by TetR-DOZI has proven to effectively reduce target gene expression [74,75,76,77,78,79,80]. PfGCN5 KD by this system resulted in only a ~50% reduction of PfGCN5 expression, likely because PfGCN5 is critical for parasite survival, and parasites may use unknown mechanisms to mitigate the reduction of PfGCN5 expression. Even under ~50% reduction, the merozoite invasion process was substantially defective, and active histone modifications (H3K9ac and H3K4me3) were profoundly reduced, particularly at the trophozoite stage. Transcriptomic analysis showed downregulation of invasion-related genes and upregulation of virulence-associated genes. Overall, the growth defects and transcriptomic changes upon PfGCN5 KD were less severe than in PfGCN5 Brd deletion.

Our previous study showed that PfGCN5 Brd deletion led to the upregulation of heterochromatin genes (e.g., virulence genes) [51], sharply contrasting with the increased gene silencing and more condensed heterochromatin structure after GCN5 KO in yeast [81]. Since Brd binds H3K9ac, we speculated that PfGCN5 Brd deletion weakened the anchoring of the PfGCN5 complex to H3K9ac, leading to the mislocalization of the PfGCN5 complex and activation of the heterochromatic genes [51]. Surprisingly, PfGCN5 KD also resulted in the upregulation of heterochromatin genes. At least eight heterochromatin-associated AP2 TFs were identified by previous reports [62,63,64,65,66], and we found that these TFs were transcriptionally altered upon PfGCN5 KD or Brd deletion. Furthermore, PfGCN5 binding motifs were found to overlap with the ones of seven heterochromatin-associated AP2 TFs. Additionally, PfGCN5 and H3K9ac were respectively enriched in Sir2A and HP1 genes, which encode two pivotal heterochromatin silencers [36,82]. Based on these findings, we hypothesize that gene activation in the heterochromatin is likely a consequence of the heterochromatin depression after PfGCN5 KD or Brd deletion.

Using the robust CUT&Tag-seq method, we successfully mapped the chromatin landscape of H3K9ac and PfGCN5 at the schizont stage, when the invasion genes are activated. GO enrichment analysis of H3K9ac and PfGCN5-regulated genes indicates that PfGCN5 is involved in the regulation of right-in-time biological processes such as merozoite invasion, while H3K9ac is involved in the regulation of more broad biological processes such as general gene transcription. Besides substantial alteration in transcriptome after Brd deletion or PfGCN5 KD, PfGCN5 was found expressed at all stages during the intraerythrocytic development cycle with a peak expression at the trophozoite stage and stress conditions (heat shock, nutrition starvation, and DHA treatment) substantially induced PfGCN5 protein expression, further validating that PfGCN5 is a master regulator in the malaria parasites [51,54]. A motif search of H3K9ac and PfGCN5 enrichment sites revealed several motifs overlapping with the binding motifs of various TFs. This is consistent with our previous finding that the PfGCN5 complex was associated with several AP2 TFs from the affinity pulldowns [51]. Similarly, some AP2 TFs were also identified in the pulldown of the GCN5 complex in *Toxoplasma gondii* [83], suggesting that the apicomplexan GCN5 complex is recruited by specific AP2 TFs.

A recent report showed that AP2-LT in vivo binding motif was not its in vitro DNA binding motif, and this in vivo binding motif overlaps with the motifs of AP2-I, AP2-P, HDP1, and SIP2, suggesting that AP2-LT is not recruited by its in vitro binding motif [65,73]. This is similar to AP2-HC, an integral member of the HPI complex, which is recruited to the chromatin by HP1, not via its in vitro DNA binding motif [64,73]. Our study showed that besides colocalization with AP2-I and AP2-P, PfGCN5 was also found colocalized with AP2-LT, further confirming that AP2-LT is an integral member of the PfGCN5 complex.

TFs recruit SAGA to the target promoters by the interactions with Tra1, a large subunit of SAGA; Tra1 is also a component of the NuA4/Tip60 complex, an MYST continuing complex [9,84,85]. Intriguingly, the homolog of Tra1 was found in the relatively conserved NuA4-like complex but not in the PfGCN5 complex in *P. falciparum* [51,86]. A genome-wide yeast-two hybrid analysis identified the interactions between AP2-LT and PfGCN5, AP2-LT and AP2-I, and AP2-LT and Pf3D7_0420300, another AP2 domain-containing protein [87]. Interestingly, AP2-LT interacts with itself, consistent with the finding that AP2-LT forms dimer [73]. It is of great interest to find which subunit of the PfGCN5 complex interacts with TFs.

## 4. Limitations of the Study

This study revealed the critical function of the GCN5 complex in a lower eukaryotic parasite that is distinct from the conserved SAGA complexes. The PfGCN5 complex appears to be a central node for regulating global gene expression by interacting with malaria parasite-specific TFs at different development stages and conditions. Although this study revealed this mechanism is different from the conserved model of SAGA complexes in gene regulation, a more in-depth study is needed to pinpoint how the PfGCN5 complex is recruited to the promoter by specific TFs in *P. falciparum*.

## 5. Materials and Methods

### 5.1. Parasite Culture

3D7 and PfGCN5 KD parasite lines were cultured under 37 °C at 5% hematocrit in 5% CO_2_, 3% O_2,_ and 92% N_2_ in the standard medium RPMI 1640 (ThermoFisher Scientific, Waltham, MA, USA) with 25 mM HEPES, 25 mM NaHCO_3_, 50 mg/L hypoxanthine, 40 mg/mL gentamicin sulfate, and 0.5% Albumax II (ThermoFisher Scientific, Waltham, MA, USA) [88]. To obtain the synchronized parasites at different asexual stages, the early-stage parasite-infected RBCs were treated with sorbitol at or incubated mature schizonts with fresh RBCs for 3–6 h [89].

### 5.2. Growth Phenotype Analysis

The growth phenotypes of the PfGCN5 KD line during the IDC were analyzed before (+aTc) and after (−aTc) KD as described [51,54]. The progression of asexual parasite development was monitored using Giemsa-stained smears every 2 h through the IDC. Parasite proliferation was measured by monitoring parasitemia starting at 0.1% rings for 7 days with medium change daily. The number of merozoites was measured by counting segmenters in each schizont. The capacities of merozoite invasion were measured using the established method [90]. Briefly, ~1 × 10^5^ schizont from PfGCN5 KD line without KD (+aTc) and with KD (−aTc) for at least 48 h were mixed with ~1 × 10^7^ fresh RBCs; 24 h later, the percentage of the ring-stage parasites developed from the invaded merozoites was measured for the calculation of invasion rate. Gametocytgenesis was induced by an established method [91,92], and the gametocytemia was measured by counting gametocytes in the Giemsa-stained thin blood smears 5 days after induction. The above experiments were conducted in at least three independent biological replications.

### 5.3. Histone PTMs

To estimate histone PTMs in the PfGCN5 KD parasite line before (+aTc) and after (−aTc) KD, histones were purified from the cultured parasites [93]. Western blotting was conducted by a standard procedure with anti-acetyl histone H3, H3K9ac (catalog no. 07-352, RRID:AB_310544, Millipore, Burlington, VT, USA), H3K14ac (catalog no. 07-353, RRID:AB_310545, Millipore, Burlington, VT, USA) anti-trimethyl histone H3, H3K4me3 (catalog no. 07-473, RRID:AB_1977252, Millipore, Burlington, VT, USA) and anti-acetyl histone H4, H4Ac (catalog no.06-598, RRID:AB_2295074, Millipore, Burlington, VT, USA) as the primary antibodies. Horseradish peroxidase-conjugated goat anti-rabbit IgGs (diluted at 1:2000) were used as the secondary antibodies. The bands were visualized using an enhanced chemiluminescence (ECL) kit (Invitrogen, ThermoFisher Scientific, Waltham, MA, USA).

### 5.4. Transcriptome Analysis

RNA-seq was performed to identify the transcriptomic changes before (+aTc) and after (−aTc) PfGCN5 KD during the IDC. ZYMO RNA purification kit was used to purify total RNA from three replicates of parasite samples at different development stages (ring, early trophozoite, late trophozoite, and schizont). The sequencing libraries were generated from the purified RNA using the KAPA Stranded mRNA Seq kit for the Illumina sequencing platform according to the manufacturer’s protocol (KAPA biosystems, Roche, Wilmington, DE, USA). Libraries were sequenced on an Illumina NextSeq 550 in the Rapid Run mode using 100 nt single read sequencing. Sequencing reads were mapped to the *P. falciparum* genome sequence (Genedb v3.1) by HISAT2 [94]. The read counts were calculated by FeatureCounts. DESeq2 was applied to analyze the differential expression [95,96], with the cutoff of *P*-adjustment of <0.01. *Transcript per million* (TPM) was used to normalize RNA-seq data. The differential expressed genes before (+aTc) and after (−aTc) PfGCN5 KD were detected based on the following two criteria. First, the P-adjustment is lower than 0.01. Second, the absolute TPM fold change is higher than 1 in all three biological replicates. The GO enrichment for differentially expressed genes before (+aTc) and after (−aTc) PfGCN5 KD was performed by GSEA as described [59,60]. The NESs were calculated to display the enrichment for a specific function or biological pathways. The raw and processed RNAseq data were deposited into NCBI GEO repository (accession number GSE284301).

### 5.5. Association Between Transcription Alterations and Chromatin Structures upon PfGCN5 KD

To analyze the association between transcriptional alterations and chromatin accessibility upon PfGCN5 KD, the ATAC-seq data, which displays ATAC peaks in 5′UTRs, was collected [70]. TSSs were assigned to the ATAC-seq peak within 1 kb. The chromatin accessibilities (ATAC-seq RPM + 0.1)/(gDNA RPM + 0.1) between differentially expressed genes upon PfGCN5 KD (the up- or downregulated genes) were calculated and RPM stands for the scaled reads per million reads. The PfHP1 enrichment data (ChIP/input) was collected from the repository [97], and the associations between the PfHP1 occupancy and transcriptional alteration upon PfGCN5 KD were calculated.

### 5.6. ChIP-seq and CUT&Tag-seq

ChIP was conducted by the established methods [51]. Briefly, RBCs infected with 3D7 parasites at the schizont stage were crosslinked with 1% paraformaldehyde, followed by neutralization with glycine (0.125 M) and then lysis with saponin. The parasites were suspended in a lysis buffer and were homogenized to release the nuclei. The nuclei were then sonicated in a shearing buffer (Microson ultrasonic cell disruptor, Farmingdale, Inc. Farmingdale, NY, USA) to shear chromatin into ~100–1000 bps. The chromatin was suspended with rabbit anti-H3K9ac (Diagenode pAb-004-050, Denvile, NJ, USA) in an incubation buffer with the addition of 20 μL of agarose beads followed by a serial of washing steps. The chromatin was then eluted by the elution buffer and reverse crosslinked. The eluted gDNA (~20 ng/sample) was then used to generate libraries using the KAPA Hyper prep kit for the Illumina sequencing platform according to the manufacturer’s protocol (KAPA Biosystems). Pooled libraries were sequenced on the Illumina NextSeq 550 instrument using 150 bp paired-end sequencing and dual indexing.

CUT&Tag was performed as described with modifications [56,58]. Briefly, parasites at the schizont stage were fixed by 1% formaldehyde and lysed by saponin, followed by suspension in a nuclear extraction buffer. A total of ~0.5 × 10^6^ nuclei were mixed with 10 μL of activated concanavalin A-coated magnetic beads (EpiCypher # 21-1401, Durham, NC, USA). The bead-bound wild-type (WT) parasite nuclei were resuspended in 50 μL Antibody150 buffer with 0.5 μg of rabbit anti-H3K9ac (Diagenode pAb-004-050) for H3K9ac CUT&Tag-seq. Rabbit IgG (Cell Signaling Technology #2729S, Danvers, MA, USA) was used as the control. The bead-bound PfGCN5::GFP nuclei were incubated with 0.5 μg of goat anti-GFP (Novus Biologicals NB100-1770, Centrnnial, CO, USA) for GCN5 CUT&Tag-seq. Meanwhile, WT 3D7 nuclei were used as the control. Nuclei were incubated with donkey anti-goat IgG (Abcam ab6885) and then resuspended in Digitonin300 buffer containing CUTANA pAG-Tn5 (EpiCypher #15-1017). Next, the nuclei were resuspended in Tagmentation buffer (10 mM MgCl_2_ in Digitonin 300) at 37 °C for 1 h, followed by washing with 50 µL TAPS buffer, quenching tagmentation with 5 µL SDS release buffer, incubating at 58 °C to release tagmented chromatin into the solution, and quenching SDS by adding 0.67% Triton-X. According to the manufacturer’s recommendations, the CUTANA High Fidelity 2× PCR Master Mix (EpiCypher #15-1018) was used to amplify the target sequences with appropriate barcoded primers, followed by capturing the DNA libraries by Kapa pure beads (Roche #KK8001, Wilmington, DE, USA). The NextSeq 550 platform was applied to sequence 150 bp paired-end reads. ChIP-seq and CUT&Tasg-seq data were submitted to NCBI GEO repository (accession number GSE284527).

### 5.7. Analyses of ChIP-seq and CUT&Tag-seq Data

The adaptor trimming and initial quality control were performed using FASTP for all CUT&Tag-seq and ChIP-seq FASTQ files [98]. The resulting reads were mapped to the *P. falciparum* 3D7 genome version 67. Peak calling was performed using ‘macs2 callpeak’ with default parameters [99]. Considering the high correlation between biological replicates of CUT&Tag-seq and to further remove noise, we only used the peaks from biological replicate 1 that had at least 80% of their regions overlapping with peaks from biological replicate 2. To test the association between epigenetic signals (H3K9ac and PfGCN5) and neighboring gene expression, we calculated the RPKM ratio between IP and control on the annotated 5′ UTR of each gene and equally classified the genes into five groups based on their quantiles. Gene expressions were plotted on a boxplot, and a correlation test (Pearson cor test) between gene expression and the median epigenetic signals in each gene group (quantile) was performed.

Motif discovery was performed using DREME on H3K9ac and GCN5 peak regions separately [72]. Randomly selected intergenic regions, matching the length distribution of H3K9ac and GCN5 peaks, were used as the negative sequence set. The size of the negative sequence set was ten times the number of peaks in each dataset. To assess the colocalization of H3K9ac or GCN5 with previously reported euchromatin-accessible signals (ATAC-seq) [70], the signals of H2A.Z [38], H3K4me3 [71], H3K9me3, AP2-I [67], AP2-LT [65,73], and AP2-P/AP2-11A [65,66], a 10 kb region centered on the summits of H3K9ac or GCN5 peaks (±5 kb) was divided into 200 bins. The ChIP-seq signal for the above-mentioned epigenetic signals was then calculated in each bin. The GO enrichment for PfGCN5- or H3K9ac-enriched genes was performed on PlasmDB (https://plasmodb.org/plasmo/) (accessed on January 2024).

### 5.8. Statistical Analysis

The results were presented as mean ± SD and regarded as significant if *p* < 0.05 by ANOVA, Fisher’s exact test, paired Mann–Whitney *U* test, or unpaired *t*-test, and the respective analysis shown in the figure legends.

## Figures and Tables

**Figure 1 cells-14-00876-f001:**
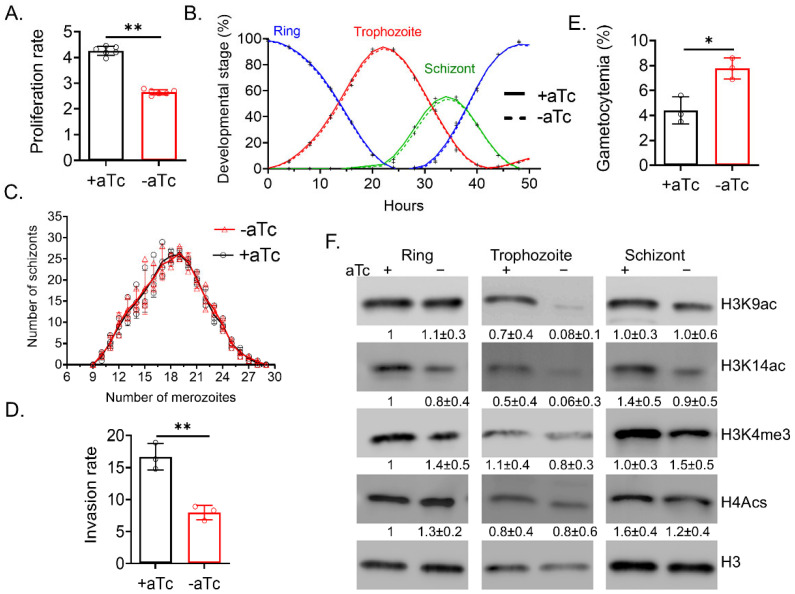
Growth and histone modification upon PfGCN5 KD. (**A**) The parasite replication rates upon PfGCN5 KD. A bar graph shows the parasite replication rate upon PfGCN5 KD (−aTc) was substantially lower than the wildtype parasites without KD (+aTc) (**: *p* < 0.01, unpaired *t*-test). Six biological replicates (*n* = 6) were conducted. (**B**) Detailed analysis of asexual parasite development through IDC by measuring percentages of stage-specific parasites (ring, trophozoite, and schizont) in the culture via Giemsa-staining every 2 h. No discernable developmental changes were identified between PfGCN5 KD (−aTc) and its control (+aTc). (**C**) Diagram displays the number of merozoites in mature PfGCN5 KD (−aTc) schizonts compared to its control (+aTc). No differences were identified between + and −aTc parasites (*n* = 3, *p* > 0.05, ANOVA). (**D**) Merozoite invasion rates after PfGCN5 KD (−aTc) was significantly decreased compared to its control (+aTc) (**: *p* < 0.01, unpaired *t*-test). (**E**) Gametogenesis upon PfGCN5 KD (−aTc) was significantly increased compared to its control (+aTc) (*n* = 3, *: *p* < 0.05, unpaired *t*-test). (**F**) The levels of active histone marks in PfGCN5 KD (−aTc) parasites compared to their controls (+aTc). Histones were purified from the parasites at the ring, trophozoite, and schizont stages. The levels of H3K9ac, H3K14ac, H3K4me3, and H4Acs were determined by Western blots after the normalization of the loading samples’ H3 protein levels (the lowest panel). Three biological replicates (*n* = 3) were conducted, and a densitometer was used to measure the band intensities. The fold changes ± standard deviations (the corresponding histone mark band vs. H3 band) were listed underneath each band. The ratios of histone mark band to H3 band in the ring stage control (+aTc) were set as 1. H3K9ac and H3K4me3 were substantially reduced at the trophozoite stage after PfGCN5 KD.

**Figure 2 cells-14-00876-f002:**
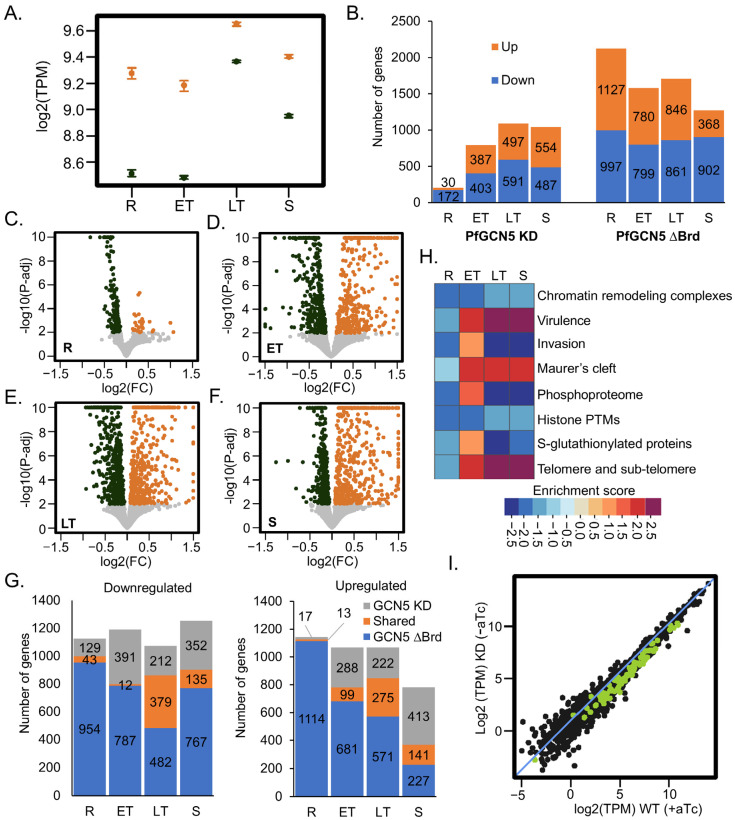
Global transcriptomic changes upon PfGCN5 KD. (**A**) A graph shows the overall transcriptional reductions of PfGCN5 after PfGCN5 KD (black) compared to their controls (orange) during IDC. R: the ring stage; ET: the early trophozoite stage; LT: the late trophozoite stage; S: the schizont stage. (**B**) Bar graphs display the total number of differentially expressed genes during IDC after PfGCN5 KD (left panel) compared to the altered genes after PfGCN5 bromodomain (Brd) deletion (∆Brd, right panel). The up- and downregulated genes are labeled in orange and blue, respectively. (**C**–**F**) Volcano plots showing altered gene expression in the rings (**C**), early trophozoites (**D**), late trophozoites (**E**), and schizonts (**F**) upon PfGCN5 KD (−aTc) compared to their controls (+aTc) from three replicates (*n* − 3) of RNA-seq. The x-axis shows log_2_ (fold change) of the express level in PfGCN5 KD parasites compared to their controls, while the y-axis displays −log_10_ (*p* values). (**G**) The downregulated (left panel) and upregulated (right panel) genes upon PfGCN5 KD and PfGCN5-∆Brd are largely overlapped at different stages. (**H**) GSEA detected the enriched functions or processes of upregulated and downregulated genes upon PfGCN5 KD. The normalized enrichment scores (NESs) were displayed by each color box. The enhanced or repressed functions upon PfGCN5 KD were depicted by the positive or negative NESs, respectively. (**I**) The dot plot shows the transcription comparison of each gene between PfGCN5 KD (−aTc) and its control (+aTc) at the late trophozoite stage. Seventy-two invasion-related genes (green dots) were generally reduced after PfGCN5 KD compared to all other genes (dark dots).

**Figure 3 cells-14-00876-f003:**
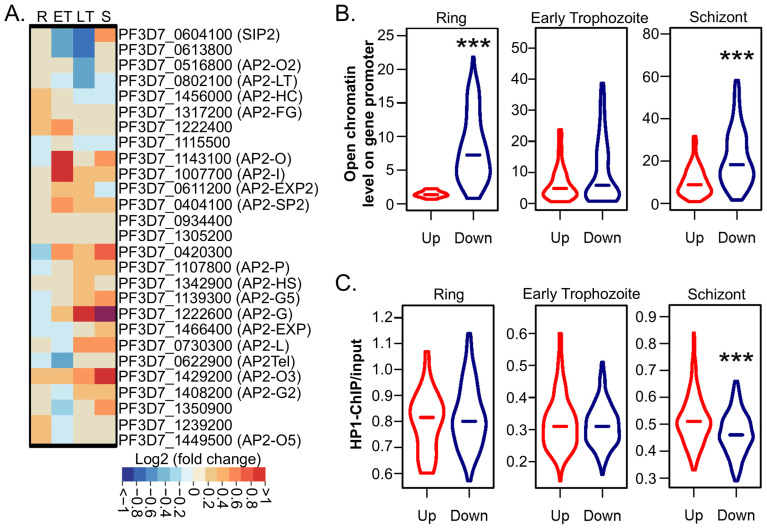
Transcriptional alteration of AP2 TFs and chromatin states upon PfGCN5 KD. (**A**) A heatmap displays transcriptional alteration of AP2 TFs upon PfGCN5 KD at different developmental stages, showing log2 (fold change between −aTc and +aTc). R: the ring stage; ET: the early trophozoite stage; LT: the late trophozoite stage; S: the schizont stage. (**B**,**C**) Violin plots show the gene transcriptional alterations upon PfGCN5 KD are negatively correlated with the accessibility of the promoters (from the ATAC-seq analysis) at all stages (**B**) but are slightly positively correlated with the heterochromatin state (represented by the HP1 occupancy) only in the rings and schizonts. (**C**). Up, upregulation; Down, downregulation. ***, *p* < 0.001 (Wilcoxon rank-sum test).

**Figure 4 cells-14-00876-f004:**
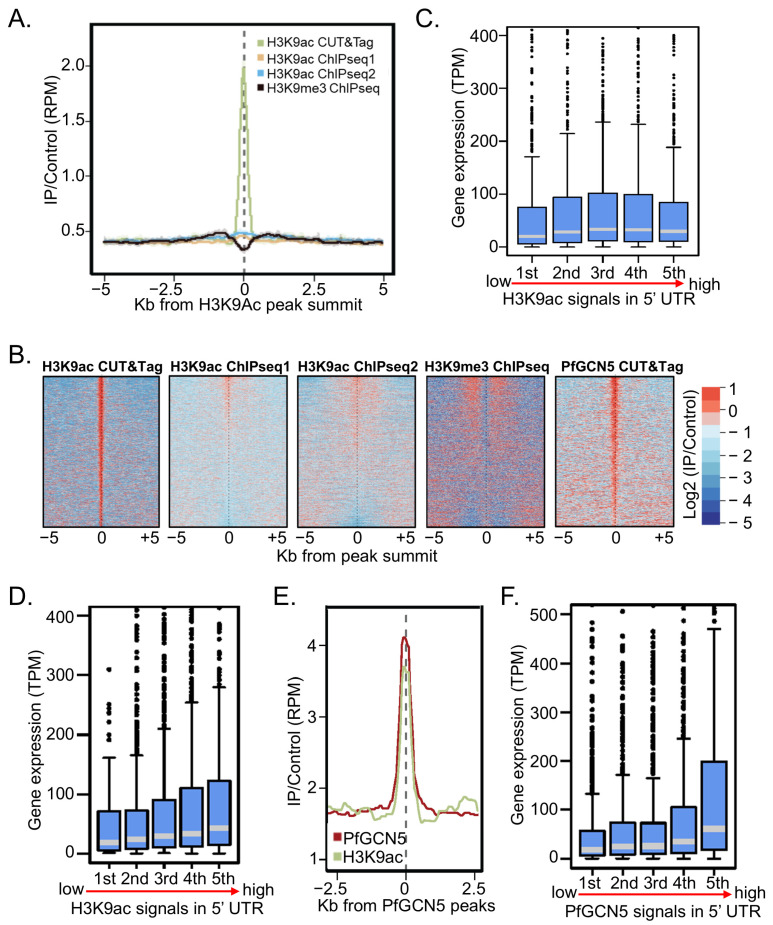
Chromatin landscape of H3K9ac and PfGCN5. (**A**) A plot shows the stronger enrichment of H3K9ac (green) by CUT&Tag-seq compared to the minor enrichment of H3K9ac in the published H3K9ac ChIP-seq (H3K9ac ChIPseq1) and our in-house ChIP-seq (H3K9ac ChIPseq2) with the same anti-H3K9ac antibodies in the WT parasites (log2 of RPKM H3K9ac/RPKM inputs or IgG). The signal of the silence mark H3K9me3 (black) is negatively correlated to H3K9ac (log2 of RPKM H3K9me3/RPKM input). (**B**) Heatmaps display H3K9ac or PfGCN5 signals from CUT&Tag-seq and/or ChIP-seq, flanked by 5 kb genome regions, showing H3K9ac and PfGCN5 enrichments compared to their controls (input or IgG) (log2 of RPKM signals/RPKM control). (**C**) A box–whisker plot shows positive correlations between gene expression and H3K9ac signals from H3K9ac ChIPseq2 at their 5′ UTRs. (**D**) A box–whisker plot shows positive correlations between gene expression and H3K9ac signals from H3K9ac CUT&Tag-seq at their 5′ UTRs. (**E**) A diagram shows the colocalization of H3K9ac (green) and PfGCN5 (red) signals from H3K9ac and PfGCN5 CUT&Tag-seq. (**F**) A box–whisker plot shows positive correlations between gene expression and PfGCN5 signals from PfGCN5 CUT&Tag-seq at their 5′ UTRs.

**Figure 5 cells-14-00876-f005:**
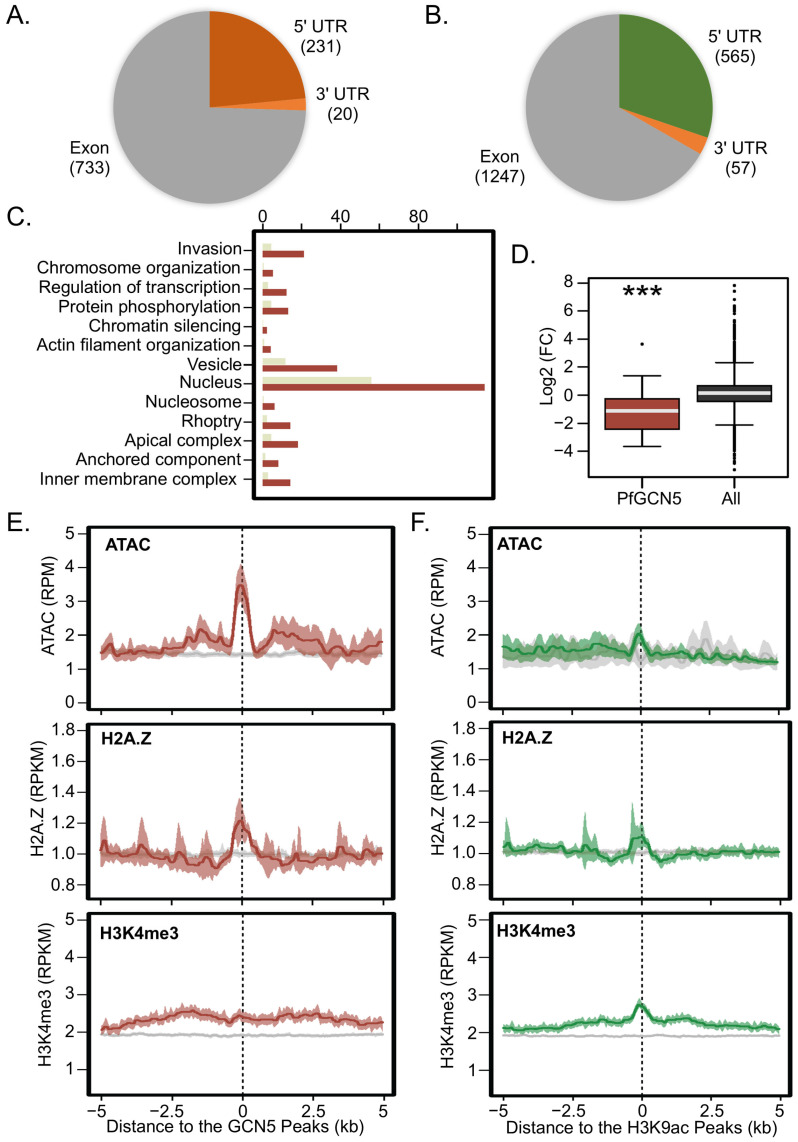
Extensive regulation by PfGCN5/H3K9ac. (**A**,**B**) Pie charts show the distribution of the identified PfGCN5 (**A**) and H3K9ac (**B**) peaks at 5′ UTRs, exon, and 3′UTRs in schizonts. (**C**) A bar graph shows GO enrichment analysis of PfGCN5-enriched genes at the schizont stage, highlighting the regulation of various biological processes by PfGCN5, such as merozoite invasion. The light green and dark red bars indicate the expected and enriched number of genes in the different pathways, respectively. (**D**) PfGCN5-enriched genes were generally downregulated compared to other genes (All) upon PfGCN5 bromodomain deletion (*** *p* < 0.001, Wilcoxon rank-sum test). (**E**) PfGCN5 signals enriched in the 5′ UTRs were colocalized with the signals from ATAC-seq, H2A.Z, and weakly colocalized with H3K4me3. (**F**) H3K9ac signals in the 5′ UTRs were colocalized with the signals from ATAC-seq, H2A.Z, and weakly colocalized with H3K4me3.

**Figure 6 cells-14-00876-f006:**
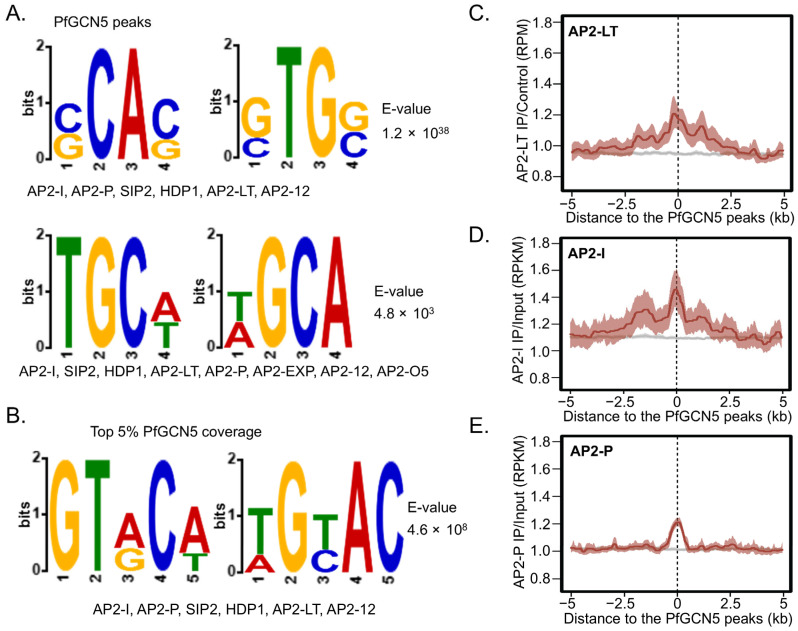
PfGCN5 chromatin binding motifs and its colocalization with AP2 TFs. (**A**) Motif enrichment analysis of PfGCN5 peaks at 5′UTRs identified several motifs overlapping with known binding motifs of certain TFs whose names were listed underneath the identified motifs. (**B**) Motif enrichment analysis of 5% PfGCN5 coverage at 5′UTRs revealed several motifs overlapping with known binding motifs of certain TFs whose names were listed underneath the identified motifs. (**C**) PfGCN5 signals at the 5′ UTRs were highly colocalized with the signals of AP2-LT (red). The gray line denotes AP2-LT signals at the sites other than the PfGCN5 peaks. (**D**,**E**) PfGCN5 signals at the 5′ UTRs were highly colocalized with the signals (red) of AP2-I (**D**) and AP2-P (**E**). The gray lines denote AP2-I (**D**) and AP2-P (**E**) signals at the sites other than the PfGCN5 peaks.

## Data Availability

RNA-Seq data were submitted to NCBI GEO repository (GSE284301). ChIP-seq and CUT&Tasg-seq data were deposited into NCBI GEO repository (GSE284527).

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
