# Peer review of "GCN5 Is a Master Regulator of Gene Expression in the Malaria Parasite Plasmodium falciparum"

_cells, 2025, doi:10.3390/cells14120876_

Round 1
Reviewer 1 Report
Comments and Suggestions for Authors
Amuza Byaruhanga Lucky et al. present a comprehensive study investigating the role of PfGCN5 in Plasmodium falciparum. By employing the CUT&Tag-seq technique, the authors successfully mapped the chromatin occupancy of PfGCN5 and its associated histone mark H3K9ac. The data demonstrate that CUT&Tag-seq offers improved sensitivity over conventional ChIP-seq for detecting the chromatin-binding landscape of pfGCN5. Mechanistically, the authors propose that multiple apetela2 (AP2)-domain-containing transcription factors—particularly AP2-LT—may serve as subunits pfGCN5 complex, guiding its recruitment to promoter regions to orchestrate broad transcriptional programs.
The study integrates extensive omics data to support its conclusions. However, several minor but significant points should be addressed to further validate the mechanistic model and enhance the manuscript:
- While pfGCN5 is described as a master regulator in the malaria parasite, it may be helpful to explore whether its expression levels or functional activity vary across different developmental stages or environmental conditions. Such information, if available, could further support its proposed role in stage-specific transcriptional regulation.
- The author claim AP2-TFs recruited pfGCN5 to these promoters. which is mechanistically intriguing. However, it would be important to assess whether the pfGCN5–AP2 complex is developmentally regulated or context-dependent. For example, does the strength or presence of this complex vary under different development stages and conditions? Such data would provide further insight into the specificity of pfGCN5-mediated transcriptional control.
- The authors report that pfGCN5 knockdown mimics the growth phenotype observed in ΔBrd mutants, suggesting that the bromodomain is functionally important. It would be valuable to explore whether the bromodomain of pfGCN5 mediates its complex formation with AP2 TFs (e.g., AP2-LT), potentially facilitating chromatin recruitment. Biochemical validation, such as co-immunoprecipitation or pulldown assays, would be helpful to support this hypothesis.
Reviewer 2 Report
Comments and Suggestions for Authors
In this study Lucky et al. continue to characterize GCN5 in Plasmodium falciparum. GCN5 is a histone acetyltransferase that is predicted to be involved in the regulation of genes associated with merozoite invasion among other virulence functions. In particular, the study authors made a conditional knockdown of GCN5 and described its phenotype including growth, invasion, development, and histone acetylation aspects. The authors also carried out comparative transcriptomic analyses, which confirmed the connection of GCN5 to invasion-related gene regulation and revealed a potential mechanism of action through interaction with AP2 transcription factors. Merozoite invasion is a critical step in Plasmodium infection and this study provides insight into that process. Therefore, the study is significant and will be of broad interest in the community. The experiments seem to be comprehensive and technically sound. Here are several areas that require clarification and improvement:
- Although the authors perform robust statistical analyses they do not report “n” for any of the experiments. This reviewer assumes that n>3 but the authors have not stated this (unless this reviewer has missed it).
- The authors report on proliferation rate of the KD parasites. What are the units? What does 2.6 vs 4.3 mean (Line 124)?
- The authors state that “…PfGCN5 KD…dramatically reduced the H3K4me level, especially at the trophozoite stage…” (Lines 162-163). Stating “especially at the trophozoite stage” implies that there is also reduction at the other stages. This is not evident in the Western blot (Fig. 1F). Furthermore, even in the trophozoite stage, this reduction doesn’t seem “dramatic.” Did the authors do statistics to analyze the differences?
- In Lines 180-181, the authors state, “…the numbers of up-and down-regulated transcripts were comparable in all developmental stages except in the ring stage…” What does “comparable” mean? Why is that relevant?
- The authors indicate that there were changes in some AP2 transcription factors, known to upregulate merozoite invasion, were themselves upregulated. Why, then, do the authors still see a decrease in merozoite invasion?
- The first sentence in Section 2.5 (lines 257-261) is very long and impossible to understand. What does it mean and why is this information relevant?
- The sentence that spans Lines 409-412 (“Enrichment analysis…”) is very long and impossible to understand. What does it mean and why is this information relevant?
- The word “drastic” or “drastically” is not conventional and is considered jargon. Please change to significant (if supported by stats) or some other word such as “marked” or “substantial.”
- The short title is not precise. It sounds like regulation of GCN5 itself and not the fact that the study is about GCN5’s ability to regulate the transcription of other genes. Maybe “Gene regulation by CGN5…” would help.
Round 2
Reviewer 1 Report
Comments and Suggestions for Authors
The authors have addressed all of my previous concerns. I have no further comments.